# Rats Lacking Dopamine Transporter Display Increased Vulnerability and Aberrant Autonomic Response to Acute Stress

**DOI:** 10.3390/biom10060842

**Published:** 2020-05-31

**Authors:** Placido Illiano, Gregory E. Bigford, Raul R. Gainetdinov, Marta Pardo

**Affiliations:** 1The Miami Project to Cure Paralysis, Department of Neurological Surgery, University of Miami Miller School of Medicine, Miami, FL 33136, USA; gbigford@med.miami.edu; 2Institute of Translational Biomedicine, St. Petersburg State University, Universitetskaya Emb. 7–9, 199034 St. Petersburg, Russia; gainetdinov.raul@gmail.com; 3St. Petersburg University Hospital, St. Petersburg State University, Universitetskaya Emb. 7–9, 199034 St. Petersburg, Russia; 4Department of Neurology and Molecular and Cellular Pharmacology, University of Miami Miller School of Medicine, Miami, FL 33136, USA

**Keywords:** dopamine transporter, stress, pituitary gland, HPA axis, autonomic response, post-traumatic stress disorder (PTSD)

## Abstract

The activity of the hypothalamus–pituitary–adrenal (HPA) axis is pivotal in homeostasis and presides the adaptative response to stress. Dopamine Transporter (DAT) plays a key role in the regulation of the HPA axis. We used young adult female DAT Knockout (KO) rats to assess the effects of DAT ablation (partial, heterozygous DAT+/-, or total, homozygous DAT-/-) on vulnerability to stress. DAT-/- rats show profound dysregulation of pituitary homeostasis, in the presence of elevated peripheral corticosterone, before and after acute restraint stress. During stress, DAT-/- rats show abnormal autonomic response at either respiratory and cardiovascular level, and delayed body temperature increase. DAT+/- rats display minor changes of hypophyseal homeostatic mechanisms. These rats display a similar pituitary activation to that of the control animals, albeit in the presence of higher release of peripheral corticosterone than DAT-/- after stress, and reduced temperature during stress. Our data indicate that DAT regulates the HPA axis at both the central and peripheral level, including autonomic function during stress. In particular, the partial deletion of DAT results in increased vulnerability to stress in female rats, which display central and peripheral alterations that are reminiscent of PTSD, and they might provide new insights in the pathophysiology of this disorder.

## 1. Introduction

Adaptative response to stress is critical in homeostasis and survival and involves the activation of the hypothalamus–pituitary–adrenal (HPA) axis. The activation of the HPA axis results in a series of neuroendocrine responses, collectively known as “the stress response”. These neuroendocrine processes have been shown to differ among healthy individuals and in persons with post-traumatic stress disorder (PTSD) [1,2], a disorder that is closely related to adverse stressful events.

Epidemiological studies have long confirmed that early trauma and stress can contribute to a number of psychiatric disorders that develop in late adolescence as well as adulthood, including, but not limited to, anxiety disorders [3], depression [4], and PTSD [5,6].

The link between involvement of the HPA axis in response to early life adverse experiences and alterations of serotonin and dopamine neurotransmission has been shown [7]. Early life aversive stressful events can be associated with mesolimbic dopamine release, which in turn affects the neurobiology of the stress response [8]. Among candidate genes that are related to stress disorders, Dopamine Transporter (DAT) disruptions have captured some attention, particularly with regards to PTSD [9,10,11,12]. Importantly, within PTSD affected individuals, there is higher incidence and prevalence of stress-related disorders in women [13,14,15,16,17,18,19,20], who show more severe symptoms and poorer treatment outcomes than men [21].

In this study, we utilize a novel rat model of hyper-dopaminergia, the DAT-Knockout (KO) Rat, to study the implication of DAT on the function of the pituitary gland in control conditions as well as in response to stress. This rat model has been recently generated and characterized [22] in order to explore behavioral repertoires that were not possible to study in the DAT-KO mouse [23]. DAT-KO rats provided further insights on the role of DAT and the neurobiology of dopaminergic neurotransmission on decision making, motivational states [24], novelty seeking [25], sexual behavior [26], and maternal care [27]. PTSD is highly prevalent in women and, therefore, we decided to direct our investigation on female rats examining three genotypes (wildtype, DAT+/+; heterozygous, DAT+/-; and, homozygous, DAT-/-). Our aim was to assess how DAT deletion affects acute restraint stress outcome in early adulthood in female rats. We focused on understanding of DAT depletion on pituitary gland structure and function, by investigating markers of cellular regulation (GM130, CyclinD1, TCF4, NCAM, HSP90), exocytotic machinery (VAMP2, SNAP25, Synaptotagmin1/2), and corticosteroid mediated feedback (Mineralocorticoid receptor (MCr), Glucocorticoid receptor (GCr)) in basal condition and in response to acute restraint stress. Furthermore, we assessed autonomic peripheral response of DAT depleted rats during restraint stress and its effects on peripheral blood markers of stress (corticosterone, IL-6). Herein, we identified that the disruption of dopaminergic neurotransmission results in aberrant pituitary gland response and abnormal autonomous function during acute stress. This work paves the way to studies aimed at characterizing the interaction between neuroendocrine and autonomic response, and DAT predisposition to stress-related disorders.

## 2. Material and Methods

### 2.1. Animals

Wistar-Han DAT-KO rats were generated, as previously described [22]. Wildtype (DAT+/+), heterozygous (DAT+/-), and homozygous (DAT-/-) female DAT-KO rats were housed in groups of 2–3, with food and water available ad libitum. The colony was maintained under standard lab conditions and under natural light cycle (12 h light/dark cycle 6 am on–6 pm off, 21 ± 1 °C, 40–70% humidity). Breeding scheme was performed as reported from our group, mating mature DAT+/- female rats of fertile age (>2 months old) with mature DAT+/- male rats [27]. All of the experiments were conducted in accordance with the National Institute of Health Guide for Care and Use of Laboratory Animals and they were approved by the University of Miami IACUC (Protocol #17-016). The 3R principle (Replacement, Reduction, and Refinement) was applied and the total number of animals used was 63. Thirty-two animals were used for experiments at the basal level (DAT+/+ *n* = 10; DAT+/- *n* = 11; DAT-/- *n*=11); 31 animals were used for acute restraint stress experiments (DAT+/+ *n* = 9; DAT+/- *n* = 11; DAT-/- *n* = 11).

### 2.2. Locomotor Activity

Locomotor activity was evaluated using Omnitech Digiscan apparatus (AccuScan Instruments, USA) under illuminated conditions. The apparatus consisted of a 40.4 cm × 40.5 × 30.3 cm plexiglass chamber, equipped with four open field monitors, each made of 16 light beams placed in the horizontal X and Y axes. The hardware detected beam breaks while the software determines the location of the animal in the locomotor box apparatus. The total distance was expressed in terms of the distance traveled by the rat, in centimeters. The stereotyped behavior (stereotypies count) was automatically quantified by the software, which measures the beam breaks patterns occurring in time intervals lower than 1 s. DAT-/- rats display spontaneous stereotyped head-weave behavior (Appendix A), which was previously reported in DAT-KO mice [28] and that can be automatically recorded by the software. Vertical activity represented the number of beam breaks on the y axes, which originated from rearing activity of the rat. The animals were individually tested for 10 min at three different time points: 35 days post-natal (PND35), 42 days post-natal (PND42), and 52 days post-natal (PND52), as briefly depicted in Figure 1A.

### 2.3. Restraint Stress

Young adult female DAT-KO rats (PND52-60) were placed in custom designed restraint cylinders (Appendix A). Briefly, a plexiglass cylinder (21/4”) allowed for body restraint. Two ring shaped plexiglass adaptors were placed around the neck and around the tail of the rat, to allow body restraint allowing head movement, in order to facilitate breathing. This setup also allowed for correct neck placement of the pulse oximeter probe. We utilized appropriate restraint tubes and body adaptors based on rat dimension, as DAT-/- have much smaller body size than the other genotypes [22]. Restraint tubes also contained aeration slots that allow for sufficient body heat dissipation (Appendix A), as reported elsewhere [29]. Restraint stress was carried out for 30 min and the rats were sacrificed by decapitation immediately after the restraint in an adjacent, separate room. Trunk blood was collected and brain was rapidly dissected on an ice-cold surface. Tissue was snap-frozen while using isopentane/dry ice, and then stored at −80 °C until use. All behavioral testing and consecutive sample collection were completed during the light phase (9 a.m.–1 p.m) with groups being randomized over the course of the day and different day sessions.

### 2.4. Autonomic Parameters Measurements

The non-invasive MouseOX instrument (STARR Life Sciences, Holliston, Ma, USA) detected the Oxygen Saturation (percentage of O_2_-saturated hemoglobin), the Heart (bpm = number of beats per minute), and Breath Rate (brpm = number of breaths per minute) and the Pulse Distention (μm = distention of the arterial blood vessels), as previously reported [30]. The instrument consists of a collar clip that can be safely placed around the neck of awake rats. The experimental rats were habituated to a mock collar clip (plastic collar with no transmission wire attached) for 10 min each day, three days before the restraint stress. The placement of the mock collar clip did not require restraint. On the test day, a wire equipped collar was placed on freely moving rats 5 min before the restraint (habituation) to obtain baseline readings (basal). After habituation, the animals were immediately restraint and the collar clip placed around the neck to provide readings throughout the duration of the session (30 min).

### 2.5. ELISA

Whole trunk blood was centrifuged at 15000 × *g* for 15 min, plasma was collected at −20 °C until analysis. The corticosterone and IL-6 concentrations were measured using commercially available enzyme-linked immunosorbent assays following manufacturer indications (Corticosterone: #ADI-900-097 ENZO; IL6: #ELR-IL6 RayBio). Each sample was analyzed in duplicate and all of the absorbance readings were performed with “Biotek ELX808” reader apparatus.

For corticosterone analysis, blood dilution was tested as per manufacturer instructions and chosen to be 35% to fall within the dynamic range of the calibration curve (Appendix A). For experiments, DAT+/+ *n* = 10; DAT+/- *n* = 10–11; DAT-/- *n* = 10–11 were used for the control group; DAT+/+ *n* = 9; DAT+/- *n* = 11; DAT-/- *n* = 11 were used for stress group.

### 2.6. Western Blotting

Western blotting experiments were performed, as previously described [31], with minor modifications. Briefly, the pituitary gland was dissected from freshly harvested brains and mechanically homogenized in extraction buffer (20 mmol/L Tris–HCl, pH 7.4, 150 mmol/L NaCl, 1% Triton-X-100, 1mmol/L EDTA, 1mmol/L EGTA) with added PhosSTOP™ (Millipore Sigma #4906837001) plus 1 tab of Complete mini, EDTA free (Roche, 11836170001). Protein concentration was measured using a Bradford assay (Bio Rad #500-0205). The protein extracts (20 μg) were separated on 4-20% Criterion(tm) TGX(tm) Precast Midi Protein Gel (Bio Rad #5671094) and transferred to nitrocellulose membranes (Bio Rad #1620112). The blots were blocked in 5% nonfat dry Milk (RPI International #M17200-500.0) in PBS-Tween20 0.05%. Primary antibodies used for Western Blot were tested for use, as per manufacturers’ instructions. Blots were incubated overnight at 4 °C with the following primary antibodies: DAT (Sigma #D6944, 1:1000); GM130 (Sigma #67295, 1:1000); TCF4 (Millipore #05-511, 1:1000); CyclinD1 (Millipore #AB1320, 1:1000); NCAM (Millipore #AB5032, 1:1000); Mineralocorticoid receptor (Millipore #AB64457, 1:1000); phospho-P44/42 MAPK (Cell Signaling #9101S, 1:1000); P44/42 MAPK (Cell Signaling #9107S, 1:1000); Synaptotagmin 1/2 (Synaptic Systems #SYSY105002, 1:1000); VAMP2 (Millipore #AB3347, 1:1000); SNAP25 (Millipore #AB5666, 1:1000); HSP90 (Sigma #H1775, 1:1000); Glucocorticoid receptor (Cell Signaling #3660S, 1:1000); β-Actin (Sigma #A2228, 1:5000); and, β-tubulin (Sigma #T8535, 1:20000). Where needed, the membranes were stripped for 10 min with Restore™ PLUS Western Blot stripping buffer (Thermo Scientific #46430), blocked, and incubated with primary antibody, as described above. After washing in PBS-Tween20 0.05%, the membranes were incubated for one hour at room temperature with the appropriate Horseradish peroxidase (HRP) conjugated secondary antibody (ECL™ Anti-Rabbit IgG Sigma #NA9340V; ECL™ Anti-Rabbit IgG Sigma #NA931, 1:2000 in PBS-Tween20 0.05%). Following secondary antibody incubations, the membranes were washed and incubated with ECL detection reagent (Thermo Fisher #34578) for 1–2 minutes. Immunoblots were acquired while using the Bio Rad ChemiDoc™ Touch systems and images were quantified using Image Lab 5.2.1 software. For each protein, the data were normalized to β-tubulin or β-actin and expressed as relative to control (DAT +/+). The absence of secondary antibodies cross-reactivity was assessed and is reported in Appendix A. The number of rats per group were as follows: nasal condition: DAT +/+ *n* = 6, DAT +/- *n* = 6, DAT -/- *n* = 4; stress condition: DAT +/+ *n* = 6, DAT +/- *n* = 6, and DAT -/- *n* = 6.

#### Western Blotting for DAT

Semi-quantitative Westerns were performed on striatal tissue using WES™ Protein Simple instrument, as previously described [32,33]. All of the samples were diluted to 0.5mg/mL (5 μL) with 0.1X sample buffer and 5X fluorescent Master Mix from ProteinSimple, then boiled at 95 °C for 5 min. Diluted samples, biotinylated ladder (5μL/well), primary (DAT sc14002, Santa Cruz Biotechnology 1:50)/secondary (10 μL/well) antibodies were loaded onto a 384-well plate for analysis by the automated capillary-based Simple Western system. The samples are separated by molecular weight through stacking and separation matrices for 25 min at 375V. Once separated, proteins were immobilized to capillary walls by applying proprietary, photoactivated capture chemistry. The capillaries were incubated with primary antibodies for 30 min, goat-anti-rabbit HRP (ProteinSimple) for 30 min, and a CCD camera with different exposure time captured the chemiluminescent signal. The signal intensities are quantified while using Compass Software and representative “virtual blot” electrophoretic image for DAT was automatically generated by the Compass Software (ProteinSimple).

### 2.7. Body Temperature Measure

Body temperature was measured at baseline and during restraint stress by means of infra-red thermal camera (Perfect Prime #IR0002). All of the experiments were conducted in temperature-controlled environments of 21±1 °C. The efficacy/utility of this non-invasive measurement of skin temperature is described in detail in the Discussion section.

### 2.8. Statistical Analysis

The locomotor behavior experiments results were analyzed with One-Way ANOVA, followed by Tukey’s *post hoc* analysis. Western blot data presented in Figure 2, Figure 3, and Appendix A show all three genotypes in basal conditions, and samples from the three genotypes were run on the same gel for each protein. One-way ANOVA with Tukey’s *post hoc* test was used for analysis of these data by marker (allowing comparisons between genotypes under basal condition)

Western Blot data presented in Figures 5–8, and Appendix A show the comparison between basal condition and stress. Each genotype (DAT+/+, DAT+/-, or DAT-/-) was separately analyzed to individually compare the effect of stress in each genotype. These two conditions for each genotype were run on individual gels. The T-test was used for analysis of these data by marker. T-tests were used to compare basal condition as compared to stress for each genotype. For basal condition DAT+/+ *n* = 6; DAT+/- *n* = 6; DAT-/- *n* = 6. For stress group, DAT+/+ *n* = 6; DAT+/- *n* = 6; DAT-/- *n* = 4.

Autonomic parameters data were analyzed while using repeated measures One-Way ANOVA followed by Tukey’s *post hoc* test; additionally, a Two-way ANOVA was performed to examine the between-group differences at each time-point.

The ELISA measurements of corticosterone and IL-6 were analyzed using Two-way ANOVA with Tukey’s *post hoc* test for multiple comparisons.

All of the presented data met the mathematical criteria for the statistical analysis chosen for analysis. GraphPad Prism 8 was used for all analyses, and the null hypothesis was rejected at the 0.05 level of significance.

## 3. Results

### 3.1. Basal Conditions

#### 3.1.1. Locomotor Behavior of Female DAT-KO Rats

Locomotor behavior was monitored during transitional age between PND 35, PND 42, and PND 52, which corresponds to the developing age from adolescence to early adulthood (Figure 1A) [34], with a significant increase in horizontal activity for DAT-/- rats when compared to DAT+/- and DAT +/+ littermates (One-way ANOVA F=(2, 6) = 80.51, *p* < 0.0001 Tukey’s *post hoc* DAT+/+ *vs* DAT+/- *p* = 0.9999; DAT+/+ *vs* DAT-/- *p* < 0.0001; DAT+/- *vs* DAT-/- *p* < 0.0001) (Figure 1B). Furthermore, stereotyped head-weave activity that was measured in the locomotor boxes was significantly higher in DAT-/- rats than both DAT+/- and DAT+/+ rats (One-way ANOVA F=(2, 6) = 166.1, *p* < 0.0001 Tukey’s *post hoc* DAT+/+ *vs* DAT+/- *p* = 0.9982; DAT+/+ *vs* DAT-/- *p* < 0.0001; DAT+/- *vs* DAT-/- *p* < 0.0001) (Figure 1C—Appendix A). Interestingly, vertical activity did not significantly differ among the three genotypes (One-way ANOVA F=(2, 6) = 0.4497, *p* = 0.6577 Tukey’s *post hoc* DAT+/+ *vs* DAT+/- *p* = 0.9796; DAT+/+ *vs* DAT-/- *p* = 0.7684; DAT+/- *vs* DAT-/- *p* = 0.6595) (Figure 1D). These results confirm that the DAT-/- female rats display a hyperactive locomotor phenotype in terms of distance traveled, as previously observed for both sexes at the same age [22]. However, while stereotyped behavior was observed, vertical rearing hyperactivity was not detected in female DAT-/- rats in early adulthood in our experimental conditions.

#### 3.1.2. DAT+/- Female Rats Display Lower Expression of DAT in the Pituitary Gland

Previous studies have shown the absence of DAT protein in striatal samples of DAT-/- rats [22]. We replicated the analysis in the striatum of young adult DAT-KO rats. We analyzed the striatal samples of DAT+/+, DAT+/-, and DAT-/- control rats, and confirmed the lack of DAT in DAT-/- rats, as well as partial reduction (~35%) of DAT levels in DAT+/- rat striatum (Appendix A). Additionally, for the purpose of this study, we then compared the pituitary DAT levels between DAT+/+ and DAT+/- rats. Our results showed a significant reduction (~3 fold) of hypophyseal DAT in DAT+/- rats (Unpaired *t* test, *p* < 0.0001 *t* = 8.109, df=10) (Appendix A).

### 3.2. DAT Ablation in Female Rats Causes Alterations of the HPA Axis

#### 3.2.1. Regulatory Elements of Pituitary Structure and Function are Affected by DAT Ablation

Mice lacking DAT display alterations in the regulation of anterior pituitary development, resulting in hypopituitarism, dwarfism, and inability to lactate [35]. In this regard, we sought to investigate the effect(s) of DAT ablation on anterior and posterior pituitary gland function in our rat model. We examined the levels of GM130 and Cyclin D1, markers that are involved in the cellular regulation of the anterior and posterior lobe of the pituitary gland, respectively. GM130 is a Golgi matrix protein [36] localized in the rat anterior pituitary gonadotrope cells [37], while Cyclin D1 is a nuclear protein that is mostly localized in the posterior lobe [38], required for the progression of cells in the proliferative G1 phase [39]. We found that the DAT-/- rats displayed significantly lower levels of GM130 than DAT+/+ controls (One-way ANOVA F = (2,13) = 6.9, *p* = 0.092 Tukey’s *post hoc* DAT+/+ *vs* DAT+/- *p* = 0.2265; DAT+/+ *vs* DAT-/- *p* = 0.0070; DAT+/- *vs* DAT-/- *p* = 0.1195) (Figure 2A,B). DAT-/- also showed a significant reduction of Cyclin D1 levels as compared to DAT+/- and DAT+/+ genotypes (One-way ANOVA F = (2,13) = 4.0, *p* = 0.0432 Tukey’s *post hoc* DAT+/+ *vs* DAT+/- *p* = 0.1685; DAT+/+ *vs* DAT-/- *p* = 0.0426; DAT+/- *vs* DAT-/- *p* = 0.5906) (Figure 2A,B). In addition, we also investigated the pituitary levels of TCF4, which is a major controlling protein of hypophyseal growth that limits the expansion of the anterior lobe [40]. The TCF4 levels were unchanged among genotypes (One-way ANOVA F = (2,13) = 1.617, *p* = 0.2360 Tukey’s *post hoc* DAT+/+ *vs* DAT+/- *p* = 0.9917; DAT+/+ *vs* DAT-/- *p* = 0.3013; DAT+/- *vs* DAT-/- *p* = 0.2571) (Appendix A). We also conducted analysis of N-Cadherin adhesion molecule (NCAM), to test whether possible alterations of architectural structure of the hypopyseal gland would occur in DAT-KO rats. The NCAM levels were not significantly affected by the ablation of DAT (One-way ANOVA F = (2,13) = 0.8055, *p* = 0.4680 Tukey’s *post hoc* DAT+/+ *vs* DAT+/- *p* = 0.6204; DAT+/+ *vs* DAT-/- *p* = 0.9421; DAT+/- *vs* DAT-/- *p* = 0.4860) (Appendix A). Taken together, these results show that homozygous DAT ablation reduces regulatory proteins of both anterior and posterior pituitary lobes, while not affecting the architecture or overall hypophyseal mechanisms of control. These data suggest that DAT-/- female rats exhibit features of hypopituitarism, consistent with their dwarf phenotype [22] and with the hypoplasia of the pituitary gland that was observed in the homozygote DAT-KO mouse [35].

#### 3.2.2. DAT Ablation Alters the Neurohypophyseal Exocytotic Machinery

We analyzed the target SNARE (t-SNARE) protein SNAP25, and vesicular SNARE (v-SNARE) protein VAMP2 to assess whether ablation of DAT affects the exocytotic machinery of the anterior pituitary [41,42,43,44]. No significant differences in VAMP2 were observed (One-way ANOVA F = (2,13) = 1.3, *p* = 0.3101 Tukey’s *post hoc* DAT+/+ *vs* DAT+/- *p =* 0.9941; DAT+/+ *vs* DAT-/- *p* = 0.3336; DAT+/- *vs* DAT-/- *p* = 0.3774), whereas SNAP25 was significantly reduced in both DAT+/- and DAT-/- genotypes (One-way ANOVA F = (2,13) = 13, *p* = 0.0007 Tukey’s *post hoc* DAT+/+ *vs* DAT+/- *p* = 0.0030; DAT+/+ *vs* DAT-/- *p* = 0.0014; DAT+/- *vs* DAT-/- *p* = 0.6803) (Figure 2C,D). In addition, we examined pituitary synaptotagmin (SynTag), which is also known to be involved in adenohypophyseal exocytosis [45]. SynTag was unchanged across genotypes (One-way ANOVA F = (2,13) = 1.508, *p* = 0.2576 Tukey’s *post hoc* DAT+/+ *vs* DAT+/- *p* = 0.2347; DAT+/+ *vs* DAT-/- *p* = 0.6023; DAT+/- *vs* DAT-/- *p* = 0.8431) (Appendix A). Taken together, these results suggest that either heterozygous and homozygous ablation of DAT interferes with the secretory function of anterior pituitary cells [46] at the pre-synaptic membrane.

#### 3.2.3. Changes in Steroid Hormones in the Pituitary Gland of Female DAT-/- Rats

We sought to measure levels of the mineralocorticoid receptor (MCr), involved in fast circulating corticosteroid negative feedback of HPA activity of the anterior pituitary in basal conditions, to further investigate the role of DAT ablation on the activity of the HPA axis [47,48]. Our results show that hypophyseal MCr was significantly increased in both DAT+/- and DAT-/- rats under basal conditions (One-way ANOVA F = (2,13)= 7.9, *p* = 0.0058 Tukey’s *post hoc* DAT+/+ *vs* DAT+/- *p* = 0.0142; DAT+/+ *vs* DAT-/- *p* = 0.0118; DAT+/- *vs* DAT-/- *p* = 0.8955) (Figure 3A,B). Additionally, we examined the levels of the glucocorticoid receptor (GCr), the main component of the negative feedback of corticosterone in the pituitary gland [48]. Our results show that, in basal condition, pituitary levels of GCr are significantly increased (two-fold) in DAT-/- rats as compared to DAT+/+ and DAT+/- (One-way ANOVA F = (2,13) = 14, *p* = 0.0005 Tukey’s *post hoc* DAT+/+ *vs* DAT+/- *p* = 0.4675; DAT+/+ *vs* DAT-/- *p* = 0.0032; DAT+/- *vs* DAT-/- *p* = 0.0005).

Mineralocorticoid and glucocorticoid receptors are known to control pituitary induction through ERK1/2-mediated transcriptional events [49,50,51]. Our results only show the basal activation of hypophyseal ERK1/2 signaling in DAT+/- rats (One-way ANOVA F = (2,12) = 5.4, *p* = 0.0217 Tukey’s *post hoc* DAT+/+ *vs* DAT+/- *p* = 0.0245; DAT+/+ *vs* DAT-/- *p* = 0.9587; DAT+/- *vs* DAT-/- *p* = 0.0665) (Figure 3C,D). Moreover, we measured the levels of the chaperone heat-shock protein 90 (HSP90), which is the main negative feedback regulator of GCr activity in the pituitary gland [52]. Our results showed that HSP90 was unchanged among genotypes (One-way ANOVA F = (2,13) = 1.359, *p* = 0.2910 Tukey’s *post hoc* DAT+/+ *vs* DAT+/- *p* = 0.9913; DAT+/+ *vs* DAT-/- p = 0.3110; DAT+/- *vs* DAT-/- *p* = 0.3625) (Appendix A) under basal conditions. Taken together, these data show profound steroidal dysregulation of the pituitary gland of DAT-/- rats, alongside a peculiar pattern of hypophyseal activation in DAT+/- rats.

#### 3.2.4. DAT Ablation Causes Alterations of Peripheral Blood Corticosterone

The female rat HPA axis displays higher activation than their male counterpart [53], which is associated with higher release of corticosterone upon acute stress [54]. Given that ablation of DAT affects hypophyseal function leading to aberrant HPA axis activation in our model, we assessed the systemic release of corticosterone, which is involved in pituitary response to stress [55]. A Two-way ANOVA for the factors genotype and stress condition yielded significant interaction (F(2,57) = 6.3; *p* = 0.0032). We found a significant effect of genotype (F(2,57) = 20; *p* < 0.0001). Across genotypes, Tukey’s post hoc analysis showed increased levels of corticosterone in DAT-/- in basal condition as compared to DAT+/+ (*p* = 0.0039) and DAT+/- (*p* < 0.0001). The effects of stress are analyzed below (please see Section 3.4.4). Taken together, these findings further support that DAT deletion alters the HPA axis within the CNS, but also alters systemic corticosterone under basal conditions.

### 3.3. Acute Restraint Stress

#### 3.3.1. Autonomic Response in DAT-KO Female Rats

Acute restraint is an unpredictable stress situation that produces a variety of autonomic responses [56], with higher activation of the HPA axis in female as compared to male rats [53]. Based on our previous findings, we decided to investigate how the alterations of the HPA axis observed would affect the central and autonomic response to acute restraint stress. In order to measure autonomic function, we used a non-invasive pulse oximeter system to measure in real-time the respiratory and cardiovascular response, and an infrared thermal camera to measure body temperature.

#### 3.3.2. Respiratory Response during Acute Restraint Stress

Arterial oxygen saturation (SpO_2_) and the number of breaths per minute (brpm) were measured to assess respiratory response. SpO_2_ analysis showed genotype significance, where the oxygen saturation was reduced in DAT-/- rats when compared to DAT+/+ and DAT+/- rats (Repeated measures One-way ANOVA genotype F = (1.0,6.2) = 18, *p* = 0.0049 Tukey’s *post hoc* DAT+/+ *vs* DAT+/- *p* = 0.6468; DAT+/+ *vs* DAT-/- *p* = 0.0130; DAT+/- *vs* DAT-/- *p* = 0.0117) (Figure 4A).

ANOVA analysis of brpm showed genotype significance (Repeated measures One-way ANOVA F = (1.4, 8.3) = 5.8, *p =* 0.0343 Tukey’s *post hoc* DAT+/+ *vs* DAT+/- *p =* 0.2536; DAT+/+ *vs* DAT-/- *p =* 0.0871; DAT+/- *vs* DAT-/- *p =* 0.1205), while no significant difference for time lapse was discovered (Figure 4B). In order to further investigate the differences between genotypes, we performed a Two-way ANOVA that showed a significant genotype effect (F(12, 195) = 2.172; *p =* 0.0002) and Tukey’s *post hoc* multiple comparisons revealed that, at 5 min after restraint, breath rate was significantly lower for DAT-/- rats when compared to DAT+/+ and DAT+/- (Tukey’s *post hoc* DAT+/+ *vs* DAT +/- *p =* 0.1119; DAT+/+ *vs* DAT-/- *p <* 0.0001; DAT+/- *vs* DAT -/- *p =* 0.0006). These data indicate that DAT-/- rats display lower breath frequency only immediately after stress restraint (5 min) and become hypoxic during the whole observational period.

#### 3.3.3. Cardiovascular Response during Acute Restraint Stress

The cardiovascular response to restraint stress was measured by pulse rate (pulse distension in µm) and heart rate (beats per minute, bpm). Pulse rate analysis showed genotype significance (Repeated measures One-way ANOVA F = (1.5, 9.0) = 5.8, *p <* 0.0001 Tukey’s *post hoc* DAT+/+ *vs* DAT+/- *p =* 0.2117; DAT+/+ *vs* DAT-/- *p <* 0.0001; DAT+/- *vs* DAT-/- *p =* 0.0001) (Figure 4C). Heart rate analysis did not show genotype significance (Repeated measures One-way ANOVA F = (1.6, 9.6) = 0.2287, *p =* 0.7518 Tukey’s *post hoc* DAT+/+ *vs* DAT+/- *p =* 0.7325; DAT+/+ *vs* DAT-/- *p =* 0.9984; DAT+/- *vs* DAT-/- *p =* 0.8378). However, repeated measures One-way ANOVA analysis showed time lapse effect (Repeated measures One-way ANOVA F=(6, 12) = 10.06, *p =* 0.0004). We performed a Two-way ANOVA that showed a significant time effect in order to further investigate the differences between time points in each genotype (F(6, 189) = 7.324; *p <* 0.0001, Figure 4D). Tukey’s multiple comparisons highlighted that, when comparing heart rate from the beginning of the restraint (5 min) to the end (30 min), DAT+/+ and DAT+/- display a significant decrease (5 min *vs* 30 min Tukey *post hoc*, *p =* 0.0163 and *p =* 0.0091, respectively), while this was not the case for DAT-/- rats (5 min *vs* 30 min Tukey *post hoc*, *p =* 0.5902).

Taken together, these data show that restraint stress in DAT -/- significantly altered both respiratory and cardiovascular response. DAT+/- rats indeed responded to restraint stress in a very similar way when compared to DAT+/+ controls in terms of overall autonomic response.

#### 3.3.4. Thermoregulation during Acute Restraint Stress

Body temperature measures showed a significant genotype effect (Repeated measures One-way ANOVA F = (1.9, 11) = 18, *p =* 0.0003 Tukey’s *post hoc* DAT+/+ *vs* DAT+/- *p =* 0.0089; DAT+/+ *vs* DAT-/- *p =* 0.3458; DAT+/- *vs* DAT-/- *p =* 0.0025) where interestingly DAT+/- rats display lower body temperature across the restraint. Repeated measures One-way ANOVA analysis also showed the time lapse effect (F=(6, 12) = 4.6, *p =* 0.0116). We performed a Two-way ANOVA that showed a significant time lapse effect in order to further investigate the differences between time points in each genotype (F(6, 188) = 4.181, *p =* 0.0006). Tukey’s *post hoc* multiple comparisons show that, following restraint stress, DAT+/+ and DAT-/- body temperature significantly increased from pre-stress baseline (DAT+/+: Tukey post hoc, *p =* 0.0100 at 5 min; *p =* 0.3615 at 10 min; *p =* 0.3907 at 15 min; *p =* 0.5048 at 20 min; *p =* 0.4834 at 25 min; *p =* 0.3243 at 30 min) (DAT-/-: Tukey post hoc, *p =* 0.4597 at 5 min; *p =* 0.1424 at 10 min; *p =* 0.0702 at 15 min; *p =* 0.0232 at 20 min; *p =* 0.0431 at 25 min; *p =* 0.0207 at 30 min), albeit at different time points (5 min and 20 through 30 min, respectively). Tukey’s *post hoc* multiple comparisons did not show any significant increase of temperature from pre-stress baseline in DAT+/-, across all time points measured for this genotype (Tukey post hoc, *p =* 0.7387 at 5 min; *p =* 0.3176 at 10 min; *p =* 0.4331 at 15 min; *p =* 0.9196 at 20 min; *p =* 0.8557 at 25 min; *p =* 0.9907 at 30 min) (Figure 5A).

Additionally, circulating levels of IL-6 were significantly increased in DAT-/- rats after stress as compared to basal, while IL-6 was unchanged in both DAT+/+ controls and DAT+/- rats after restraint stress (Two-way ANOVA genotype x stress interaction F=(2, 56) 3.432, *p =* 0.0393; genotype condition (F(1, 56) = 7.693; *p =* 0.0075) and stress condition (F(2, 56) = 2.261; *p =* 0.1137). Tukey’s *post hoc* comparison showed DAT+/+ basal *vs* DAT+/+ stress *p =* 0.9970; DAT+/- basal *vs* DAT+/- stress *p =* 0.9903; DAT-/- basal *vs* DAT-/- stress *p =* 0.0051) (Figure 5B). Finally, we analyzed the levels of HSP90 after stress exposure. While its expression did not change in DAT+/+ controls after restraint, two-fold decrease was measured in DAT+/- whereas a three-fold increase was observed for DAT-/- genotype (Unpaired *t*-test, two-tailed, *p =* 0.3199 DAT+/+ control *vs* stress; *p =* 0.0141 DAT+/- control *vs* stress; *p <* 0.0001 DAT-/- control *vs* stress) (Figure 5C,D).

These data demonstrate that restraint elicited a hyperthermia effect that was peculiar in DAT+/- and DAT-/- rats. While DAT+/- rats display lower body temperature during restraint accompanied by a reduction in pituitary levels of HSP90, DAT-/- displayed hyperthermia during restraint, albeit at later time points than DAT+/+ controls (Figure 5A). This occurs alongside an increase in circulating levels of pro-inflammatory cytokine IL-6 (Figure 5B) and increases in pituitary levels of HSP90 (Figure 5C,D). These data point towards a restraint-mediated hypothermia in DAT+/- rats, while indicating an aberrant central and peripheral hyperthermic restrain stress response of DAT-/- rats.

### 3.4. HPA Axis Response to Acute Restraint Stress

Herein, we observed alterations of the HPA axis in both DAT+/- and DAT-/- in basal conditions, along with variations of the autonomic response to restraint stress. Therefore, we proceeded to investigate how the restraint stress affected pituitary function and blood cytokines in these rats.

#### 3.4.1. Functional and Structural Components of the Hypophysis are Affected by Restraint Stress

Protein levels of the anterior lobe marker GM130 were differently affected among genotypes after restraint stress. While DAT+/+ control rats showed a significant decrease (~2-fold), GM130 was not significantly altered after stress in DAT+/- rats. On the contrary, DAT-/- rats showed a significant increase (~2-fold) in GM130 levels after restraint stress (Unpaired t-test, two-tailed, *p =* 0.0001 DAT+/+ control *vs* stress; *p =* 0.0977 DAT+/- control *vs* stress; *p =* 0.0161 DAT-/- control *vs* stress, (Figure 6A,B). Moreover, the levels of the posterior lobe marker Cyclin D1 was significantly reduced after restraint stress in both DAT+/+ and DAT+/- rats, while its levels significantly increased (~2-fold) in the pituitary gland of DAT-/- rats (Unpaired t-test, two-tailed, *p =* 0.0352 DAT+/+ control *vs* stress; *p =* 0.0114 DAT+/- control *vs* stress; *p =* 0.0035 DAT-/- control *vs* stress, Figure 6C,D).

We then analyzed whether TCF4 and NCAM were affected by stress. In both cases, the levels of these proteins were significantly higher in DAT-/- rats after restraint, with a ~2-fold increase for TCF4 (Unpaired t-test, two-tailed, *p =* 0.6662 DAT+/+ control *vs* stress; *p =* 0.5829 DAT+/- control *vs* stress; *p =* 0.0087 DAT-/- control *vs* stress) and ~3-fold increase for NCAM, respectively (Unpaired t-test, two-tailed, *p =* 0.0986 DAT+/+ control *vs* stress; *p =* 0.4514 DAT+/- control *vs* stress; *p =* 0.0001 DAT-/- control *vs* stress, Figure 6E–H). Restraint stress did not affect the levels of these proteins in both DAT+/+ and DAT +/- rats. Taken together, these results provide a first indication that restraint increases all of the measured hypophyseal regulatory elements in DAT-/- rats, while only mildly affecting those in each DAT+/+ and DAT+/- genotype.

#### 3.4.2. Exocytotic Machinery of Pituitary Gland Is Altered after Restraint

Evidence from our experiments indicating alterations of the HPA axis, and vulnerability to stress of the pituitary function and structure elements, prompted us to investigate whether restraint could affect the exocytotic machinery of DAT-KO rats. Restraint stress significantly increased VAMP2 in all genotypes when compared to their non-restrained control (Figure 7A,B). Separated post hoc analyses reveal robust significant effects in DAT+/+ and DAT+/- (Unpaired *t*-test, two-tailed, *p =* 0.0047 DAT+/+ control *vs* stress; *p =* 0.0166 DAT+/- control *vs* stress), effects that have an even stronger significance level in DAT-/- rats (*p =* 0.0017 DAT-/- control *vs* stress). Interestingly, this effect was smaller in DAT+/- rats, where VAMP2 levels were 1.5-fold higher, compared to a two-fold increase observed in both DAT +/+ and DAT-/- groups (Unpaired t-test, two-tailed, *p =* 0.0047 DAT+/+ control *vs* stress; *p =* 0.0166 DAT+/- control *vs* stress; *p =* 0.0017 DAT-/- control *vs* stress). Pituitary levels of SNAP25 showed a significant four-fold increase in DAT-/- rats, while it was unchanged after stress in both DAT+/+ controls and DAT+/- (Unpaired t-test, two-tailed, *p =* 0.2534 DAT+/+ control *vs* stress; *p =* 0.1588 DAT+/- control *vs* stress; *p <* 0.0001 DAT-/- control *vs* stress, Figure 7C,D). Pituitary levels of Synaptotagmin 1/2 (SynTag) were significantly increased in DAT+/+ and DAT+/- rats after stress, while its levels were unchanged in DAT-/- rats (Unpaired t-test, two-tailed, *p =* 0.0003 DAT+/+ control *vs* stress; *p =* 0.0004 DAT+/- control *vs* stress; *p =* 0.3514 DAT-/- control *vs* stress, Figure 7E,F). Overall, these results indicate that the stress effects on the hypophyseal exocytotic machinery are similar in DAT +/+ controls and DAT+/- rats, whereas the response of DAT-/- to restraint is abnormal for SNAP25 and SynTag.

#### 3.4.3. Restraint Stress Alters Steroid Hormones Receptor Levels and ERK1/2 in the Hypophysis of Female DAT-KO Rats

The pituitary levels of MCr were unchanged in DAT+/+ and DAT+/- rats after 30 min of restraint stress, while a two-fold increase was observed in the DAT-/- genotype (Unpaired t-test, two-tailed, *p =* 0.1774 DAT+/+ control *vs* stress; *p =* 0.7550 DAT+/- control *vs* stress; *p =* 0.0069 DAT-/- control *vs* stress, Figure 8A,B). The disruption of the GCr mediated negative feedback of stress response was observed in DAT-/- rats, where the levels of this protein were significantly reduced (Figure 8C,D). Restraint stress did not affect GCr in either DAT+/+ and DAT+/- genotypes (Unpaired t-test, two-tailed, *p =* 0.6441 DAT+/+ control *vs* stress; *p =* 0.1660 DAT+/- control *vs* stress; *p =* 0.0008 DAT-/- control *vs* stress, Figure 8C,D). Moreover, the expression of the downstream signaling protein ERK1/2 was unchanged by stress in the hypophysis of DAT+/+ and DAT+/- rats. Interestingly, the phosphorylation of ERK1/2 was significantly decreased after stress in the pituitary of DAT-/- rats (Unpaired t-test, two-tailed, *p =* 0.1668 DAT+/+ control *vs* stress; *p =* 0.2439 DAT+/- control *vs* stress; *p =* 0.0150 DAT-/- control *vs* stress, Figure 7E,F). Taken together, these results show changes in the steroidal signaling of the hypophysis of DAT-KO female rats, with different effects that are dependent on the heterozygosis and homozygosis of DAT gene.

#### 3.4.4. Restraint Stress Differentially Increases Circulating Corticosterone in DAT-KO Rats

The corticosterone levels were measured in basal condition in all three genotypes (rats not exposed to stress, data presented in Section 3.2.4) and in a second group of rats from each genotype exposed to restraint stress (Two-way ANOVA F=(2, 57) 6.3, *p =* 0.0032). Restraint stress induced significant increases in blood corticosterone levels in all three genotypes as compared to basal levels (F(1,57) = 70.17; *p <* 0.0001). Noticeably, the highest increase in circulating corticosterone (2.87-fold) was observed in DAT+/- rats (Tukey’s *post hoc* DAT+/+ basal *vs* DAT+/+ stress *p =* 0.0049; DAT+/- basal *vs* DAT+/- stress *p <* 0.0001; DAT-/- basal *vs* DAT-/- stress *p =* 0.0415). Moreover, the levels of corticosterone in DAT-/- rats after stress were significantly higher than those of DAT+/+ rats after stress (DAT+/+ *vs* DAT+/- *p =* 0.8023; DAT+/+ *vs* DAT-/- *p =* 0.0342; DAT+/- *vs* DAT-/- *p =* 0.4415) (Figure 3E).

## 4. Discussion

DAT-KO rats represent a novel tool to investigate neuropsychiatric diseases that are related to dysfunctions of dopamine-related disorders [57,58]. Since their generation [22], this animal model has provided new insights in the study of disease states characterized by aberrant central dopamine [24,25]. The rat model shows higher complexity than DAT-KO mouse [23] with a wider behavioral repertoire and ease of manipulation [59]. A major difference between the DAT-KO mouse and the rat is the percentage of survival of the homozygote population. While it has been observed that ~30% homozygote DAT-KO mice die because of neurodegeneration of GABAergic medium spiny neurons [31,60], there is currently no evidence of increased mortality or neurodegeneration in the homozygote DAT-KO rats (DAT-/-), at least up to 4–6 months of age [22]. This has allowed our group and several other teams to conduct behavioral, neurochemical, and pharmacological investigations [22,24,25,34,61] that could not be performed in the knockout mouse model.

Recent studies have focused on DAT hypofunction, demonstrating that the heterozygote deletion of the transporter, both in mice [62] and in rats [24], is relevant to neuropsychiatric disorders, like attention-deficit, hyperactivity disorder (ADHD), and vulnerability to stress. Cinque and coworkers have recently shown that female heterozygous rats (DAT+/-) born from DAT+/-:DAT+/- breeding display hyperlocomotion response upon exposure to combined isolation-housing stress [24]. The authors assess that the phenotype could be the outcome of altered maternal care provided by DAT+/- dams that ultimately affects behavior in adult females. However, it is possible that the response to stress of DAT+/- adult females could also be due to intrinsic alterations of the HPA axis.

Analysis of the estrous cycle through vaginal smearing was not performed because of the high unpredictability of both DAT+/- and DAT-/- response to handling stressors. This technique, albeit non-invasive, requires hand-restraining. This type of restraint, along with other manipulation techniques, such as tail vein injection and decapitation, are known to affect cardiovascular response up to three hours in female rats [63]. Furthermore, from our experience, the handling of hyperactive DAT-/- rats might require even longer restraining time in order to properly collect vaginal secretion, thus possibly affecting the outcome of subsequent acute restraint stress.

Studies on both mouse and rat DAT knockout model have shown that DAT deletion modulates the HPA axis [35,61]. In rats, the activation of HPA axis upon stressful stimuli is much greater in females than males [54] and the dysregulation of the HPA axis has been associated with post-traumatic stress and anxiety disorders in rats and human, with a higher prevalence for female sex in both species [53,64,65].

In basal conditions, our data did confirm hyperlocomotion behavior that was previously seen on mixed population of DAT-/- rats [22]. Moreover, herein, we report stereotyped head-weave behavior for DAT-/- female rats with adolescence onset, thus highlighting a possible role of aberrant dopaminergic neurotransmission in the early onset of manic/compulsive behaviors [66,67]. These data are in line with previous observation on DAT-/- rats, which have shown to display oral stereotypies [24]. Our repeated testing design, across early and late adolescence, also allowed for the measurement of anxiety in terms of vertical activity, which is a mixed rearing behavior that includes supported (against wall of locomotor box) and unsupported rearing (no contact with walls of the arena) [68]. Our results clearly display that DAT-/- rats do not show higher rearing during the adolescence–early adulthood. Albeit our repeated testing paradigm was different than the one used to characterize DAT-KO rat model, it is possible that the higher vertical activity observed at 1 month and 1.5 months of age in homozygote rats (equivalent of PND 35 and PND 52 in this study) might reflect the hyperactivity of male rats, as the results were obtained from both sexes in the previous study [22].

In the present study, we investigated the impact of dopaminergic transmission on regulatory events of the hypophysis in DAT-/- female rats in basal conditions and after restraint stress. The observed reduction of control mechanisms of the anterior and posterior lobe of the pituitary gland supports the hypothesis that hypopituitarism, a feature previously observed in DAT-KO mice [35], which could contribute to the dwarf phenotype displayed, affects DAT-/- rats. Indeed, hypopituitarism is deemed being majorly responsible for the absence of sufficient release of growth hormone and prolactin to support normal body growth and lactation in DAT-KO mice [35]. Further studies are needed to elucidate the role of DAT ablation on the regulation and possible depression of hypohpyseal growth hormone-releasing hormone and the consequent the dwarfism that DAT-/- rats also exhibit. Importantly, hyperactivity could affect calories consumption and body fat composition and, therefore, contribute to the smaller phenotype that was observed for this genotype. Furthermore, we believe that alterations in pituitary function might also affect prolactin release and, therefore, possibly account for the altered maternal care displayed by DAT+/- females [24]. Further studies aimed at characterizing the neuroendocrine profile of DAT-KO rats are needed to elucidate how central hyperdopaminergia affects the endocrine system.

Interestingly, under basal conditions, DAT+/- and DAT-/- rats both display lower pituitary levels of the t-SNARE protein SNAP25. The reduction observed in DAT+/- clearly shows that abnormalities in the exocytotic machinery are dependent on the increase of dopaminergic tone originating from the hypothalamus, and not merely due to the hypopituitarism of DAT-/- rats. Taken together, these data show that central dopaminergic signaling controls pituitary synaptic release and, therefore, plays a key role in the regulation of the HPA axis in basal conditions and the absence of external stressors.

Presently, little is known regarding how hyperdopaminergia affects the interplay between pituitary glucocorticoid (GCr) and mineralocorticoid (MCr) receptors. MCr binds aldosterone and corticosterone with similar affinity, while the GCr has less affinity for corticosterone than MCr [69]. Our data show that the blood levels of corticosterone are increased in DAT-/- rats. These data highlight that DAT-/- rats display, in basal condition, chronically elevated glucocorticoids and exacerbated pituitary sensitivity to circulating steroids, which might result in deleterious effects on many systems, chief amongst them the cardiovascular system [69]. Full DAT deletion results in serious insult to the HPA axis, specifically in the pituitary function, which, in turn, increases central and peripheral vulnerability to stress. Further studies are needed to clarify how hyperdopaminergia affects the different pituitary cell populations and consequently their neuroendocrine function in hormonal synthesis and release.

Importantly, ERK signaling appears to be increased only in DAT+/- under basal conditions, thus highlighting a unique profiling of pituitary function modulated by DAT hypofunction. Previous data have shown anomalies in the HPA axis of DAT+/- rats, as the norepinephrine content was higher in the hypothalamus and hippocampus only in this genotype [25]. Reportedly, norepinephrine regulates the activity of the secretagogue corticotropin-releasing hormone (CRH) neurons in the paraventricular nucleus (PVN) of the hypothalamus [70]. Alterations of norepinephrine release at this point of the HPA axis might then underlie the peculiar pituitary response to circulating steroids and higher levels of MCr in DAT+/- rats. Interestingly, we also report that the heterozygote deletion of DAT also affects the increase of corticosterone levels after stress, leading to a ~3-fold increase when compared to basal conditions in this genotype, accounting for the highest increase among the three genotypes. DAT+/- rats display peculiar hypophyseal activity in basal conditions and increased levels of corticosterone immediately after stress, which resembles the findings of increased cortisol in women that are affected by PTSD [71].

For this study, we used a non-invasive pulse oximetry to record autonomic response because the implant of a catheter would have required anesthesia, which in itself can be a stressor [72]. Similarly, to measure full-body temperature, we opted for an infra-red camera. We are aware that measurement taken with thermal camera do not correspond to internal temperature measurements taken with anal thermometers or abdominal probe. Nevertheless, our data from DAT+/+ controls show same extent of temperature increase measured with abdominal probe (+1.5 °C) in female rats of equivalent age restrained with a similar apparatus, 10–15 min during restraint [29]. Furthermore, continuous temperature monitoring through repeated rectal probe insertions might also be considered stressful in rodents [73] and add to an unknown extent to the restraint stress. The infra-red camera also allowed for us to obtain overall body temperature measurements, instead of tail skin temperature, which is currently widely employed as non-invasive technique [74]. Our data confirm a role of dopamine involvement in the autonomic control of respiration during stress, as DAT-/- animals display a reduction of respiratory function and severe hypoxia upon restraint. The attenuation of respiratory response to hypoxia is mediated by the sensitivity of the carotid body that is mainly dopaminergic [75]. As DAT-/- rats are hyper-dopaminergic [22], the increase of synaptic dopamine might account for the decreased respiratory rate that was observed immediately after restraint. This, in turn, might contribute to prolonged hypoxia development, due to a dysfunctional respiratory network. Our results are in line with previous work describing the link between respiratory psychophysiology and stress neuroendocrinology via the HPA axis [76]. In short, processing of perceptual information—in this case restraint stress—can have profound effects on HPA axis activity and, subsequently, on respiratory parameters. In this paradigm, hyperdopaminergia might act as a physiological ‘ceiling’, wherein a normal respiratory stress response cannot be made. These illustrate multiple mechanisms by which DAT KO might affect basal respiratory networks and the response to stress.

Furthermore, the general increase in the cardiovascular response and rise in body temperature observed upon restraint might also arise from alteration in the HPA-axis. It is well understood that long-term cardiovascular adaptations to activity/exercise include a greater stroke volume (where pulse distention is considered a proxy) and a reduced heart rate, resulting in greater mechanical efficiency of the circulatory system [77]. Interestingly, in the hyperactive DAT-/-, we report similar trends in these parameters. Of importance, the early heart rate response to stress is thwarted, but it remains higher as time elapses, which suggests that the acute response to stress and the recovery mechanisms are dysregulated. We also observe that the pulse distention remains elevated in DAT-/- when compared to the other genotypes, thus supporting autonomic dysregulation of the cardiovascular response to stress. Sustained hyperthermia itself might also account for the dramatic increase of HSP90 protein in the pituitary gland of DAT-/- rats after stress, along with increased levels of peripheral IL-6. Our findings support previous literature showing that the hypothalamus has been implicated in the local neurotransmission that controls neuroendocrine and autonomic response to restraint stress, with changes in arterial pressure and heart rate [78]. The dramatic central, peripheral, and autonomous response to stress reveals the predicted vulnerability to stress of this animal model [27]. The pituitary gland appeared to be hyper-responsive to restraint in DAT-/- rats, as the majority of its regulatory elements, exocytotic machinery, and corticosteroid feedback mechanisms were 2- to 4-fold increased. Further demonstration of the heightened responsivity of the HPA axis resides in the elevation of serum corticosterone levels in the DAT-/- genotype, which is known to mediate the restraint stress response in rats [79]. Nonetheless, it is important to also consider that the elevation of corticosterone could concurrently result from the severe hypoxia induced by restraint, as previously reported [55].

Noticeably, DAT+/- rats show marked lower body temperature during restraint stress, which is also accompanied by a reduction in the levels of HSP90 in the pituitary gland and a marked increase of peripheral corticosterone. While the majority of the regulatory elements, exocytotic machinery components, and elements of corticosteroid-mediated negative feedback undergo mild changes that resemble those of control DAT+/+ rats, partial ablation of DAT could affect response to stress in a ‘paradoxical’ opposite way when compared to DAT-/- rats. Further studies are needed in order to elucidate how DAT heterozygosis might affect response to stress at both HPA-axis level and in the periphery.

Finally, note the potential utility of DAT-KO rats from a translational perspective, with respect to the most recent criteria to claim the validity of animal models of psychiatric disorders [80]. Our data strongly support that partial and total deletion of DAT exert differential effects on pituitary function, susceptibility to stress, and peripheral response of the HPA axis. Multiple lines of evidence have shown a link between early life trauma and PTSD [5,6,81]. In addition, the dysregulation of the HPA axis and increased acoustic startle response (previously linked to hyperdopaminergia) are recognized as endophenotypes of this disorder, where DAT represents one of the candidate genes involved in PTSD pathophysiology [9,10,11,12]. The work presented herein showed that DAT+/- display both face and predictive validity for PTSD susceptibility, specifically in regards to circulating corticosterone as a biomarker, along with alterations in pituitary function as an outcome of stress response. Observations from previous studies also confirmed the mechanistic validity of this model for PTSD susceptibility, as both cognition (asocial phenotype) and neurobiology of the HPA axis are severely affected [61]. Our results give support to the use of DAT+/- to elucidate the relevant biological mechanisms underlying PTSD pathophysiology.

In conclusion, our work highlights the role of dopaminergic regulation of pituitary, autonomic, and peripheral function in stress-free conditions and after acute stress in young adult female rats, shedding light on mechanism of the response to stress that involve the HPA-axis, with relevance to pituitary function. Further, we believe the use of DAT-KO rats as a novel, complex, and predictive model of susceptibility to neuropsychiatric diseases, which could provide further insight in the neurobiology and pathophysiology of these disorders, paving the way for the development of new therapeutic approaches and strategies.

## Figures and Tables

**Figure 1 biomolecules-10-00842-f001:**
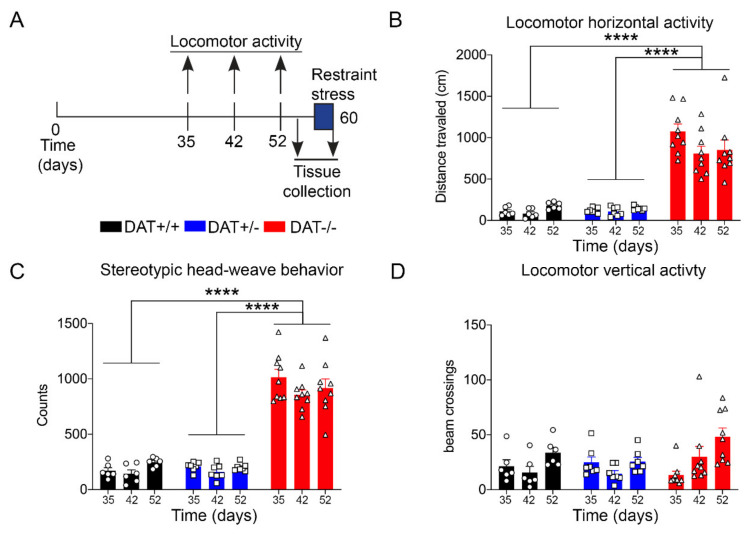
Locomotor activity in Wild-type control (Dopamine Transporter (DAT) +/+), Heterozygous (DAT +/-), and Homozygous (DAT -/-) rats. (**A**) Schematic diagram for locomotor activity, restraint stress and tissue collection. Genetic ablation of DAT leads to elevation in (**B**) Horizontal activity, and (**C**) Stereotypic behavior, without alterations in (**D**) Vertical activity, on postnatal day (PND) 35, 42, and 52, (*n*= 7–8 per group). Data were expressed as mean ± SEM in the 0–10 min period. *****p* < 0.0001.

**Figure 2 biomolecules-10-00842-f002:**
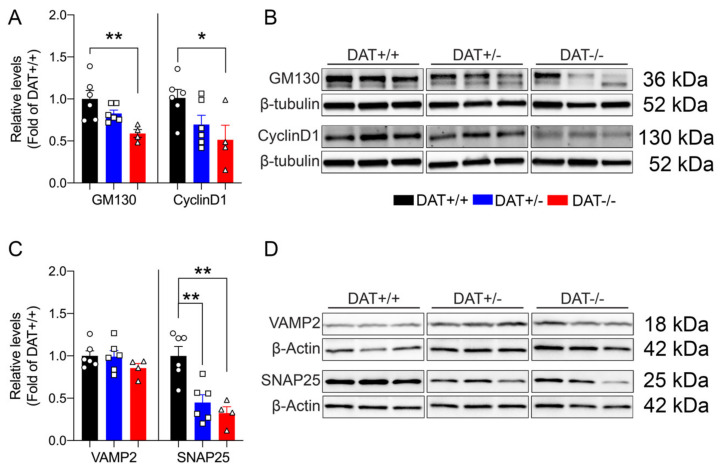
Effects of DAT ablation on the cellular regulation and exocytotic machinery of the pituitary gland. (**A**) Full DAT ablation (DAT -/-) reduced protein levels of GM130 and CyclinD1. (**C**) DAT ablation did not alter VAMP2, but reduced protein levels of SNAP25 for both DAT+/- and DAT-/- rats. (**B**,**D**) Immunoblots from representative rats in each group are shown (PND60). Data were normalized to β-tubulin or β-actin, relative to control (DAT +/+). Data were expressed as mean ± SEM (*n* = 4–6), **p* < 0.05, ***p* < 0.01.

**Figure 3 biomolecules-10-00842-f003:**
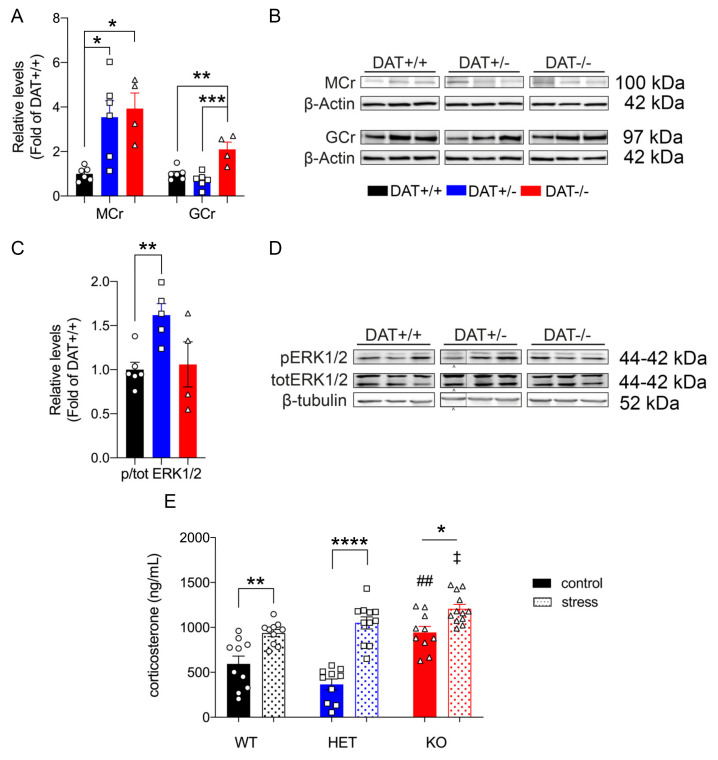
Steroid hormones and ERK1/2 changes in the hypophysis of female DAT-KO rats. (**A**) Mineralocorticoid receptor (MCr) levels were increased in both DAT+/- and DAT-/- female rats, while Glucocorticoid receptor (GCr) levels were increased only in DAT-/- animals. (**C**) DAT +/- rats showed increase ERK1/2 protein levels. (**B**,**D**) Immunoblots from representative rats in each group are shown (PND60). (**E**) DAT ablation causes alterations of circulating blood levels of corticosterone in basal conditions and after stress. Data were normalized to β-tubulin or β-actin, relative to control (DAT +/+). Data were expressed as mean ± SEM (*n* = 4–6). **p* < 0.05, ***p* < 0.01, ****p* < 0.001, *****p* < 0.0001. ^##^*p* < 0.01 DAT-/- *vs* DAT+/+ and DAT+/- in basal conditions. ^‡^*p* < 0.05 DAT+/+ stress *vs* DAT-/- stress. ^ indicates same-gel/genotype/condition non-contiguous bands.

**Figure 4 biomolecules-10-00842-f004:**
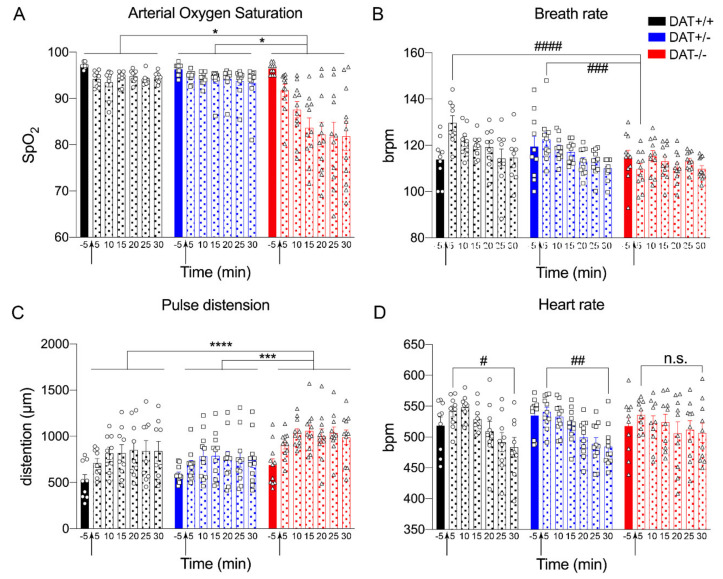
Autonomic response in DAT-KO rats exposed to restraint stress. (**A**) Restraint stress severely reduced arterial oxygen saturation (SpO_2_) in DAT -/- rats. (**B**) Breath rate (breaths per minute - brpm). (**C**) Pulse distension (µm) was also affected by restraint stress in DAT-/- rats. (**D**) Heart rate (beats per minute – bpm). Arrows indicate start of the restraint. ‘-5′ indicates measure of parameters 5 min prior the initiation of restraint (basal). The data were expressed as mean±SEM (*n* = 9–11), **p <* 0.05, ***p <* 0.01, ****p <* 0.001, *****p <* 0.0001 compared to DAT+/+ controls; #*p <* 0.05, ##*p <* 0.01, ###*p*
*<* 0.001, ####*p <* 0.0001 between time points indicated.

**Figure 5 biomolecules-10-00842-f005:**
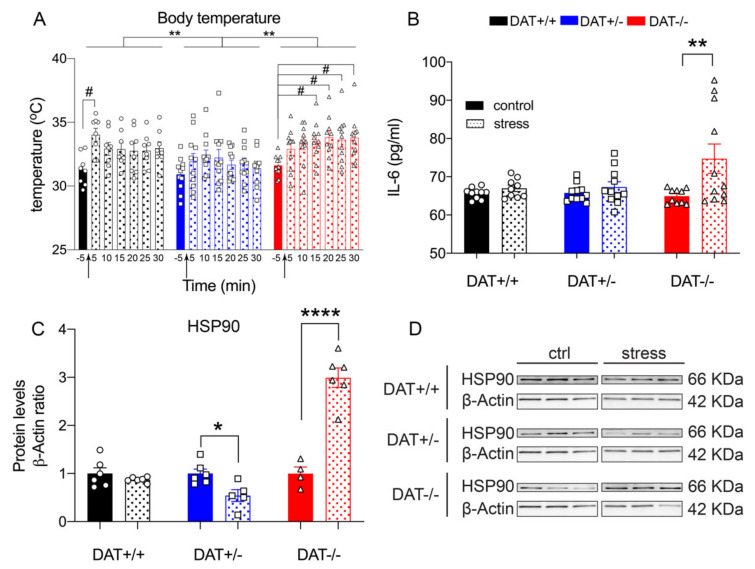
Peripheral and central effects of restraint stress on temperature, IL-6 and HSP90 levels in DAT-KO rats (**A**) Body temperature of DAT+/- rats is lower than that of DAT+/+ and DAT+/- rats upon restraint. (**B**) Circulating levels of IL-6 cytokine were increased in DAT-/- rats after restraint stress. (**C**) Restraint stress exposure significantly reduced HSP90 in DAT+/- rats; however, the stress exposure increased HSP90 protein levels in DAT-/- female rats. (**D**) Immunoblots from representative rats in each group are shown. Data were normalized to β-actin, relative to control (DAT +/+). For each genotype, data were relative to basal (non-stress) conditions and normalized to β-actin. Data were expressed as mean ±SEM (*n* = 4–6). **p <* 0.05; ***p <* 0.01 *****p <* 0.0001, #*p <* 0.05.

**Figure 6 biomolecules-10-00842-f006:**
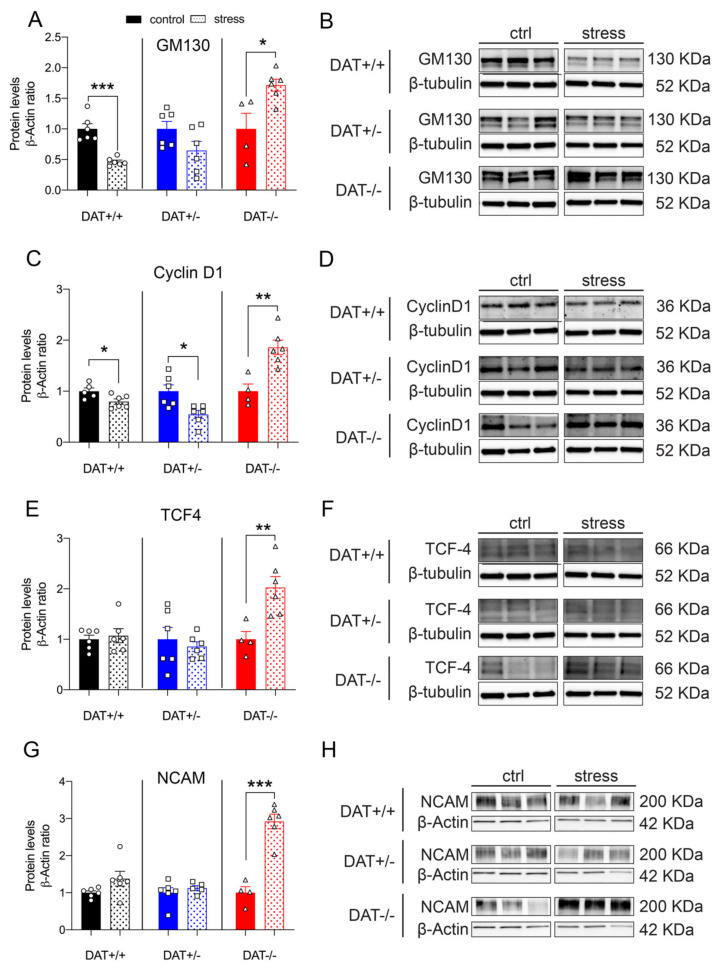
Effects of restraint stress on functional and structural components of the pituitary gland. (**A**) Exposure to restraint stress significantly reduced levels of GM130 in control rats (DAT +/+); however, no changes were induced in DAT +/- rats while significant increases were found in DAT-/- rats. (**C**) Cyclin D1 protein levels were significantly reduced after stress in DAT+/+ and DAT +/- rats; however, the effects of stress in full DAT ablated rats were opposite compared to their own basal levels. Restraint stress increased (**E**) TCF4 and (**G**) N-Cadherin adhesion molecule (NCAM) protein levels in DAT-/- rats but did not alter levels in DAT+/+ or DAT+/- rats. (**B**,**D**,**F**,**H**) Immunoblots from representative rats in each group are shown. For each genotype, data were expressed as relative to basal (non-stress) conditions. Data were normalized to β-tubulin or β-actin, relative to control (DAT +/+). Data are expressed as mean ±SEM (*n* = 4–6), **p <* 0.05, ***p <* 0.01, ****p <* 0.001.

**Figure 7 biomolecules-10-00842-f007:**
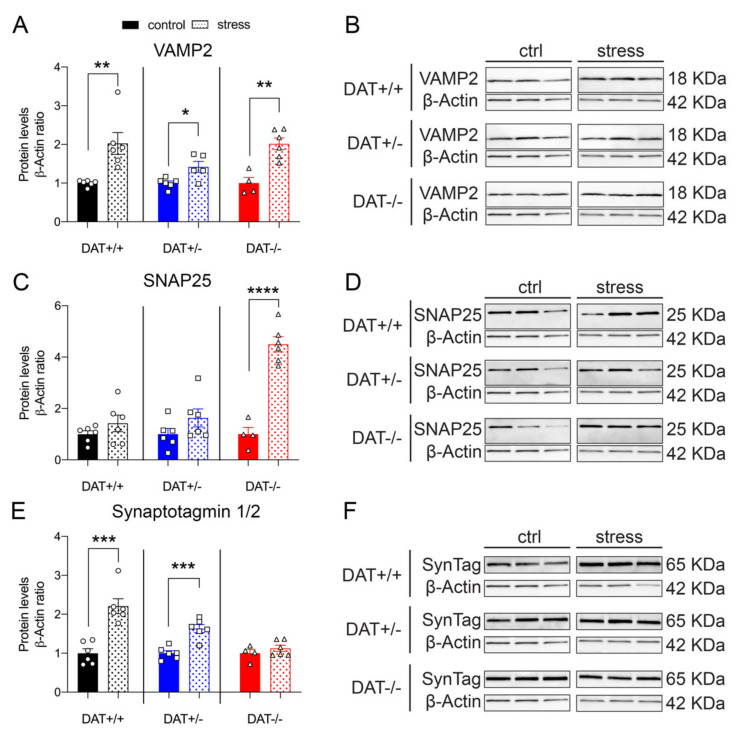
Restraint stress effects on the exocytotic machinery of the pituitary gland in female rats lacking DAT. (**A**) Exposure to restraint stress significantly increased VAMP2 in all genotype conditions. (**C**) Restraint stress increased SNAP25 protein levels only in DAT-/- rats. (**E**) Increased levels of synaptotagmin 1/2 protein were found in DAT+/+ and DAT +/- rats after stress, but no changes were detected in DAT-/- rats. (**B**,**D**,**F**) Immunoblots from representative rats in each group are shown. Data were normalized to β-actin. For each genotype, data were expressed as relative to basal (non-stress) conditions. Data are expressed as mean ±SEM (*n* = 4–6), **p <* 0.05, ***p <* 0.01, ****p <* 0.001. *****p <* 0.0001.

**Figure 8 biomolecules-10-00842-f008:**
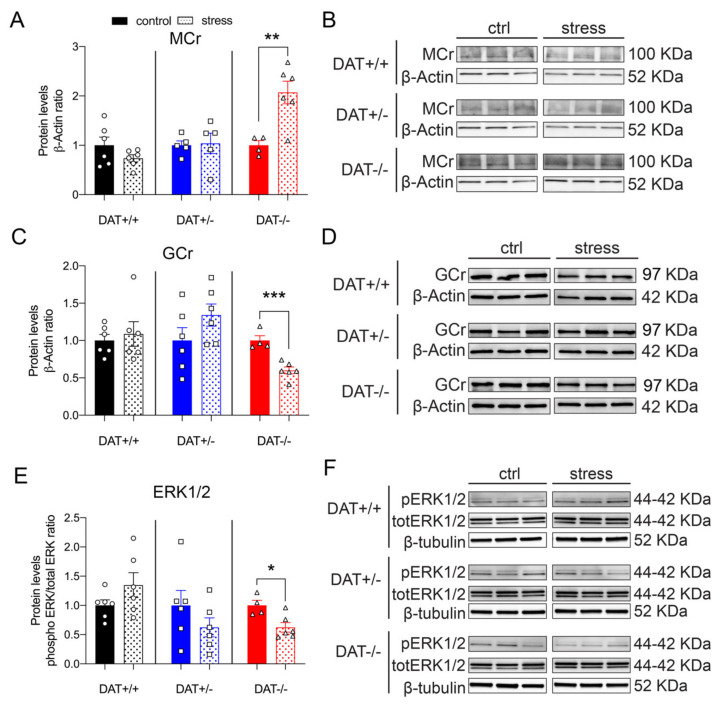
Restraint stress effects on steroid hormones and dopamine D2 receptor signaling in the hypophysis of female DAT-KO rats. (**A**) Exposure to restraint stress significantly increased MCr in DAT-/- rats. (**C**) GCr protein levels were significantly reduced after stress in DAT-/-. (**E**) The effects of restraint stress on ERK1/2 protein levels were significantly reduced only in DAT-/- rats. (**B**,**D**,**F**) Immunoblots from representative rats in each group are shown. Data were normalized to β-actin or β-tubulin. For each genotype, data were expressed as relative to basal (non-stress) conditions. Data are expressed as mean ±SEM (*n* = 4–6), **p <* 0.05, ***p <* 0.01, ****p <* 0.001.

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
