# Peer review of "Rats Lacking Dopamine Transporter Display Increased Vulnerability and Aberrant Autonomic Response to Acute Stress"

_biomolecules, 2020, doi:10.3390/biom10060842_

Round 1

Reviewer 1 Report

The manuscript by Illiano et al., .uses a recently described DAT-KO line to explore alterations in the HPA axis. The data are interesting and the conclusions that the DAT plays a role in HPA function are somewhat novel. Moreover, this report begins to detail some mechanisms by which the DAT may regulate pituitary function. However, there are major concerns with an inappropriate use of statistical analyses, lack of description of normalization processes, and the validity of some of the conclusions.

  1. A One-Way ANOVA, not a Two-Way ANOVA should be performed for the locomotor behavior readouts in Figure 1.

  1. A western blot showing loss of DAT in the KO and if the DAT heterozygote has an ~50% reduction in expression should be shown. It is possible that the Het expression may be normal or not as reduced. Ideally, this should be done in the pituitary; however, if DAT expression is too low there, showing in a DAT enriched location (e.g. striatum) would be nice. While these mice were previously made, validation is important.

  1. The definition of stereotypies in the methods section is a little vague. Also, there is a typo in this part that makes it a little confusing. Specifically, the manuscript states “The stereotypies count represented the number of repeated beam breaks patters occurring during the observation period.” I believe it is supposed to say patterns, but even this is a little vague. How much repetition defines a pattern?

  1. Since the N’s are ~7-11 per group, please show graphs as scatter and mean rather than bars to see all of the individual data points.

  1. Please note if the western blots for the different genotypes were all run on the same gel. They have to be in order to compare expression. The way the figures are laid out, it may be interpreted that they were not run on the same gel. I assume that showing different boxes for the 3 different groups was done because there were intervening lanes on the same gel, but make sure to mention that normalization to the DAT+/+ condition was performed on the respective gel. The methods used for quantification and normalization of western blots (not just acquisition of the signal) should be added to the methods.

  1. The statistics comparing the western blots in figures 2 and 3 are inappropriate. They should be an ANOVA followed by a multiplicity-corrected post-hoc test (Tukey etc.,). You cannot do an unpaired t-test without running an ANOVA first and it is more appropriate to perform a multiplicity-corrected post-hoc test. Using an unpaired t-test of non-restrained and restrained is not the best comparison for the experiments in figures 5-7; however, it may be the only way to compare if all 6 groups for each N were not run on the same gel. If possible it would be better to normalize to DAT+/+ and then do a Two-Way ANOVA (genotype and restraint) followed by post hoc test. Whatever is done, needs to be better described in methods as mentioned above.

  1. The blot for pERK1/2/total ERK does not look increased at all. The blot does not match the quantifications for the DAT+/-.

  1. For the SpO2 and BPM experiments, a One-Way, repeated measures ANOVA should be performed for the experiments in Figure 4.

  1. Body temperature studies are a little problematic. Normal rat body temperature is much warmer than what the infra-red camera is detecting (usually ~37). This should be stated as a caveat as it is not truly an internal body temperature. If they were measuring 31, rectally, they would be very sick.

  1. The statement “Interestingly, this effect was smallest in DAT+/‐ rats, where VAMP2 expression was 1.5‐fold higher, compared to a 2‐fold increase observed in both DAT +/+ and DAT‐/‐ groups.” Is not appropriate. A TWO-WAY ANOVA was not run and so just because the absolute value is different, you do not know if the difference is statistically significant. The statement “A 2.87‐fold increase was observed in DAT+/‐ rats, which was significantly higher than the increase observed for both DAT+/+ and DAT‐/‐ rats (Unpaired t‐test, two‐tailed, p<0.0001 and p<0.0001 respectively)” in regards to corticosterone is also inappropriate. You cannot do a t-test in this way. You have to do a Two-way ANOVA of the groups to see if the increase is different. Also, for the ELISA, it is unclear why they are normalizing their data. Shouldn’t they report absolute quantitations across the samples by comparing to a standard curve and report and perform statistics on those numbers? They should not be normalizing to the non-restrained condition for these assays and should report and analyze absolute concentrations.

  1. In the Discussion, the authors state “Nevertheless, we report herein locomotor stereotyped behavior for DAT‐/‐ female rats with adolescence onset, thus highlighting a possible role of aberrant dopaminergic neurotransmission in the early onset of manic/compulsive behaviors” This is a stretch for interpreting their data because their definition of stereotypies is not well described. Are there alterations in grooming? They say there are no vertical movement changes which are also associated with stereotyped behaviors.

  1. Is smaller size in the rats just due to decrease body weight because they are hyperactive. A better discussion of this should be done. Were their lengths measured? Was body composition performed?

  1. Given that D2 activity decreases prolactin production in the anterior pituitary, it is surprising that the authors did not measure circulating prolactin or prolactin in the lactotrophs as greater dopaminergic tone would be predicted to decrease prolactin secretion. They mention lactation in the introduction, but that is all.

  1. The statement “DAT+/‐ rats display hypophyseal activity and decrease of corticosterone in basal conditions which resembles findings in human PTSD pathophysiology [2, 65]” should be clarified to state “which resembles findings for cortisol levels in human PTSD pathophysiology” since humans don’t have corticosterone.

Reviewer 2 Report

This manuscript by Placido Illiano and colleagues characterizes the effect of homo- and heterozygous deletion of the dopamine transporter on the HPA axis activity and the response to acute restraint stress.

The findings are in part conclusive and potentially interesting, but there are several issues to be treated in order to increase the value of the work.

Major critical remarks:

  1. Experimental design: Authors performed acute restraint stress with a thick-wall plastic cylinder that was not perforated like a grid. This interfered with the heat loss of the rat and added to an unknown extent to the stress-induced hyperthermia of the rat. Therefore, these results are not reliable.
  2. Efficacy of stress exposure. The absolute CORT level is relatively low, in the controls, that only shows 1.59-times increase upon restraint. This is not convincing, about the stress efficacy. Taking the other physiological variables in consideration, animals might have experienced some stress, but the low CORT suggests some technical problems with CORT measurement (see point 3). What was the duration of restraint stress exposure? Is it possible that the CORT titer already returned to the baseline at the time of blood sampling?
  3. Inappropriate use of methods. Authors performed ELISA test to determine the corticosterone (CORT) titer in the blood. Comparing the results of CORT levels with the manual of the ELISA kit used (https://www.enzolifesciences.com/ADI-900-097/corticosterone-elisa-kit/) the values are out of the range the test may detect. Therefore, CORT findings are not reliable.
  4. Missing controls. Authors used many antibodies for Western blotting, but nothing is written about the specificity tests. This makes the results of those tests at least questionable. Also to these, the MCR Western blot band is very weak, hardly visible.
  5. Gel photos and graphs do not fit always: see Fig 3A vs. B. Here, for MCr the beta actin is low in +/+ rats, while for GCr the beta actin is low for the -/-. Here the question arises if the housekeeping protein was stable as that might be regulated by stress (see ref PMID: 18722514). Also here, the 1.5 times rise of pERK in 3C cannot be recognized in panel D.
  6. Statistics. Authors performed various statistical tests in this experiment. In some cases three groups are compared pairwise with t tests, which not correct. ANOVA tests and Tukey post hoc comparisons would be required, which was used for some of the datasets. For example, Authors claim at ln. 339 that "stress affects hypophyseal regulatory elements similarly in DAT+/+ and DAT+/‐, and in a significant and opposite manner in DAT‐/‐ rats." But nothing is written about genotype x stress interaction and post hoc comparisons which would support this statement.
  7. Authors do not claim that their datasets met the mathematical criteria of t test and ANOVA.
  8. Still to the statistics, in many cases (for example datasets in Fig. 4.) repeated measures ANOVA would be required.
  9. Authors use the term "expression" in a too wide meaning. Relative protein content was determined by immunoblot measurements, as it shows the amount of the protein found in the sample that is the net result of dynamic changes in production by translation as well as protein breakdown and/or release. The gene expression wasn't determined here as that would have required mRNA work. Also here, Authors even say that the corticosterone would be expressed. (ln 249.) This is steroid hormone that is produced or released but not "expressed".
  10. Authors measure many proteins, but one would expect to read something in the introduction about the rationale of these measurements. Some data one can find in the Methods part, but this is not the right place to explain what does support that the measurement of these variables was actually necessary.

Minor points:

  1. ln. 64 "These works pave the way..."
  2. Authors may explain their test for stereotypic behavior more detailed.
  3. Please correct the grammar error in subheading title 3.2.3.

Reviewer 3 Report

The Ms by Iliano and Colleagues reports a well structured, performed rigorously and interesting study that addresses the issue of the role of DAT in female controls and DAT heterozygous and homozygous female rats on a number of behavioral (at different stages) and physiological, structural/molecular parameters linked to the HPA axis both under normal and upon presentation of a stressor.  The results, clear and well presented, show that  this "hyperdopaminergic condition" signficantly impacts on the hyphyseal and HPA axis central and peripheral targets and functions.  Perhaps in this regard the Authors should  dedicate a paragraph to discuss the limitations of the model also in light of some differences also shown in the Ms between homozygous and heterozygous as well as on the potential role and the lack of control over the estrus cycle (though permissible, given the complexity of the study).  A critical issue that I believe is relevant Authors address refers to the definition of restraint stress as mild (besides I could not see for how long  rats were restrained (if I do infer exactly 30 minutes would not be a mild stressor) nor which condition was adopted to "control" the manipulation related to the restraint itself). Moreover, Authors should clearly state also which rats (were they the same or others?) and how many rats were used to collect  "autonomic responses" after restraint, in particular with respect to the statment that rats were sacrificed by decapitation immediately after the restraint.

Round 2

Reviewer 1 Report

I commend the authors on their completely addressing the critiques and believe the manuscript is acceptable for publication.

Author Response

We thank the reviewer for the revision of our manuscript

Reviewer 2 Report

Authors performed extensive revisions, but some major points remain to be addressed.

Experimental design. Authors did not describe at what time did they stress the animals. The paper Authors cite in their answer to this remark (McGivern et al., 2009 Physol) describes that the restraint stress effect on core temperature is a function of circadian cycle phase.  CORT levels are also sensitive to the daily rhythm.

Efficacy of stress exposure. 5000pg/ml CORT value is very-very low. Normal control male rats have a CORT level of 50-100 nmol/l (=17300-34600pg/ml). The CORT titer in their blood increases up to 300-800 nmol/l in restraint stress exposure. Also the cited paper (Scherer et al., 2011 Physol Behav) reports 400-600ng/ml CORT values upon 30 mins restraint, which is 400.000-600.000pg/ml. Authors report the 1/40 of this value upon stress. The validation of the CORT values with an alternative method would be essential.

The Western blot requires positive and negative control lysate tests, endogenous positive control if possible, and omission of primary antibody control. It is unclear what do Authors mean with “Immunostaining was performed using antibodies…”.

Authors ignored the advice to avoid the term expression when it is about relative protein content in all figures, however they mostly corrected the text body on this issue.

Minor points:

Please correct errors at:

ln 135. Western blot experiments were

ln. 172. capillaries were

ln. 227. hypophyseal

ln. 343. brpm vs bprm

Reviewer 3 Report

I'm satisfied with the changes provided. I suggest to change the adjective "traumatic" associated to stress at the end of the introduction.

Author Response

We thank Reviewer #3 for the revision of our manuscript. We have substituted the adjective “traumatic” with “acute” (Page 2, line 70) to be consistent with the rest of the manuscript.

Round 3

Reviewer 2 Report

Authors did an excellent job this time, and addressed all my tachnical and content-related concerns with this revision. No additional points to be corrected were found, therefore, I suggest the acceptance of this manuscript version for publication.